# Vegetation health monitoring based on sub-daily sap flow variability

Anna T. Schackow<sup>1,2</sup>, Susan C. Steele-Dunne<sup>3</sup>, David T. Milodowski<sup>4</sup>, Jean-Marc Limousin<sup>5</sup>, and Ana Bastos<sup>1,2</sup>

Correspondence: Anna Schackow (a.schackow@studserv.uni-leipzig.de), Ana Bastos (ana.bastos@uni-leipzig.de)

**Abstract.** The terrestrial biosphere plays a critical role in regulating carbon and water fluxes. Rising global temperatures increase atmospheric dryness, which in turn raises atmospheric water demand on vegetation and places. Some plants regulate transpiration losses by closing stomata, at the cost of reduced carbon uptake. Quantifying stomatal regulation and detecting early onset of vegetation stress at large scales remains a challenge.

Sap flow in stems responds to water potential gradients between the roots and the atmosphere, and therefore provides a window into transpiration and stomatal regulation. Here, we demonstrate how variations in the diurnal cycle of sub-daily sap flow as a function of vapor pressure deficit (VPD) measurements can elucidate the different levels of plant hydraulic stress. We derive two metrics based on sub-daily responses of sap flow to VPD: the morning sensitivity, given by the slope of the bi-variate relationship, and the area of the diurnal sap flow – VPD curve. We find that the morning slope, is positively associated with top (0-30cm) soil moisture, i.e., soil water availability. The area of the diurnal cycle, characterizing the degree of daily hysteresis between sap flow and VPD, is sensitive to temperature and soil moisture, increasing with sap flow downregulation before peak VPD.

While in situ sensors can provide continuous sap flow data, we consider the potential to estimate these descriptors of the diurnal cycle using temporally sparse data. In particular, as sap flow is connected to changes in water storage, which can be estimated using microwave remote sensing, we examine the degree to which the slope and area can be estimated for several acquisition strategies that vary in terms of the numbers of observations and acquisition times. We argue that sub-daily microwave observations, with at least three sub-daily overpasses could be used to characterize the hysteresis and enable improved monitoring of biosphere dynamics and vegetation health.

<sup>&</sup>lt;sup>1</sup>Institute for Earth System Science and Remote Sensing, Leipzig University, Germany

<sup>&</sup>lt;sup>2</sup>Max Planck Institute for Biogeochemistry, Dept. of Biogeochemical Integration, Jena, Germany.

<sup>&</sup>lt;sup>3</sup>Department of Geoscience and Remote Sensing, Delft University of Technology, the Netherlands

<sup>&</sup>lt;sup>4</sup>School of GeoSciences, University of Edinburgh, Edinburgh UK

<sup>&</sup>lt;sup>5</sup>4 CEFE, Univ Montpellier, CNRS, EPHE, IRD, Montpellier, France

#### 1 Introduction

**Figure 1.** Conceptual representation of the diurnal variation in sap flow (SF, blue) and vegetation water content (VWC, orange). Hysteresis in the relationship between both quantities and vapor pressure deficit depends on environmental conditions and stress. It is hypothesized that characterizing this hysteresis with sub-daily observations at key times could provide insight into vegetation health, stress state and stress response.

- Rising CO<sub>2</sub> levels drive rising temperatures, which in turn intensify the atmospheric demand for water (Vicente-Serrano et al., 2022a). This leads to reduced water availability, particularly in semi-arid regions, and heightens the risk of plant hydraulic stress or even hydraulic failure (Allen et al., 2015; Vicente-Serrano et al., 2022b). Climate change thus poses a significant threat to global vegetation health (Hartmann et al., 2022), raising serious concerns about ecosystem stability and the planet's ability to regulate climate (Bustamante et al., 2023).
- Plant-water dynamics is a fundamental indicator of plant functioning, connecting the carbon, water and energy cycles. Water loss through evaporation in the stomata (transpiration) reduces the leaf water potential, inducing the flow of sap from the soil through the root-stem-leaf continuum and supporting the transport of nutrients through the xylem (Hammond et al., 2021). Regulation of stomatal aperture (stomatal conductance) allows plants to respond to changes in environmental conditions in order to preserve the continuity of the plant water column (Jarvis and McNaughton, 1986a) and avoid embolism (Hammond et al., 2021), at time-scales from seconds to seasons or longer (Konings et al., 2021).

Ecosystems have evolved varying strategies to cope with high atmospheric water demand and limited soil moisture availability. Maintaining high stomatal conductance through periods of high atmospheric water demand may increase gross carbon uptake and therefore promote faster growth, but increased rates of evapotranspiration may risk embolism and deplete water reserves if dry conditions persist (McDowell et al., 2022). Downregulation of transpiration through stomata reduces water loss, thus preserving water reserves, but also reduces the diffusion of CO<sub>2</sub> into the leaf, thus coming at the expense of reduced carbon uptake in the short term, and the availability of carbon for growth and plant maintenance, in the long run (Hammond et al., 2021). The outcome of these different strategies is complex and depends on multiple factors, such as drought characteristics, environmental factors, ecosystem memory and compounding effects (Cranko Page et al., 2022; Peltier and Ogle, 2023; Bastos et al., 2021) and biotic interactions (Seidl et al., 2017). Tracking the responses in vegetation water content and fluxes to changing atmospheric conditions is crucial for understanding the vitality of global ecosystems and assessing their potential risk of mortality (Preisler et al., 2021; Hammond et al., 2021; Landsberg et al., 2017; Rowland et al., 2015).

Vegetation undergoes daily fluctuations in water status driven by atmospheric demand, for example the high transpiration demand (VPD peak) around solar noon (Figure 1). During the growing season, plant water potential typically ranges from relatively high values at pre-dawn, when soil water is generally not limiting and incoming radiation is low, to lower values during solar noon and the afternoon, particularly on clear-sky days when light and evaporative demand are high. These diel fluctuations are part of the normal operating conditions of xylem water transport and are regulated by stomatal responses, which balance carbon uptake and transpiration-driven water loss (Jarvis and McNaughton, 1986b). Healthy plants with sufficient soil water recharge their water storage overnight, when transpiration demand is low (Forster). Increasing transpiration through the morning decreases leaf water potential, which drives a corresponding increase in SF. Stomatal regulation may decouple transpiration, and consequently sap flow (SF), from atmospheric conditions (Franks et al.). In contrast, periods of prolonged drought can deplete root-zone soil moisture, reducing overnight recharge of vegetation water storage and lowering pre-dawn water content and plant water potential, leading to a certain level of stress (Davis and Mooney, 1986; Limousin et al., 2009, 2010b). We therefore hypothesise that certain characteristics of the sub-daily response of SF to VPD can help fingerprint periods and drivers of hydraulic strategies of vegetation, disentangling top-down (i.e. atmospheric demand) and bottom-up drivers (i.e. soil moisture supply) of hydraulic limitation and water use strategies.

As illustrated in Figure 1, VWC and SF usually show pronounced daily cycles, driven by atmospheric water demand (represented by vapor pressure deficit, VPD), incoming radiation, temperature, and plant hydraulic functioning. Under low-stress conditions 1(b and d), such as during spring, VWC and soil moisture are high. From pre-dawn (here represented as 6 am) to solar noon (here represented as 12 pm), VWC declines while SF increases steeply with VPD, peaking before or together with VPD. The rates of declining VWC or increasing SF indicate efficient water transport, if aligned with VPD versus active stomatal regulation, when hysteresis occurs in the diurnal cycle. SF can decrease despite rising VPD, indicating regulation in response to cumulative water loss. From mid-night to pre-dawn, SF approaches zero, and plants begin to recharge overnight, returning VWC to pre-dawn levels during the night. Under water-limited conditions, stomatal regulation reduces SF sensitivity to VPD. Morning increases in SF are muted, and VWC changes more slowly, especially in the afternoon. Pre-dawn VWC is lower due to incomplete overnight recharge. Nighttime VWC values remain low if water is not sufficiently refilled, and stress

effects are more pronounced around mid-day. Variations in the sub-daily relationships between VWC, SF, and VPD therefore provide insight into plant water regulation strategies.

Limitations in soil water supply are therefore expected to be indicated by lower sap flow rates that increase more slowly with atmospheric demand during the day as stomatal regulation restricts further transpiration losses, resulting in hysteresis loops with a lower VPD-SF gradient during the morning (Figure 1(c)). Additionally, factors like embolism, insect or fungal infestations, and disease can impair plant hydraulic conductivity (Torres-Ruiz et al.), leading to hysteresis stress signatures that may be decoupled from meteorological drivers. As sub-daily dynamics of sap flow (Figure 1 (b and c)) and vegetation water content (Figure 1 (d and e)) reflects the plant's ability to respond to abiotic stressors, both can provide an indicator of plant health and vitality. Furthermore, as water transport and stomatal regulation are near-instantaneous responses to stress conditions (Choat et al., 2018), hydraulic hysteresis signatures potentially provide early-warning indicators that may significantly precede optically visible signs of vegetation decline (Hammond et al., 2021). We thus hypothesize that the capacity to characterize this hysteresis through sub-daily observations of SF or VWC, or proxies for them, could provide a window into vegetation health and enable early detection of vegetation water stress before structural decline or canopy-level changes become detectable.

Currently, processes associated with plant functioning such as plant productivity and evapotranspiration are monitored at sub-daily temporal resolution in networks of sites such as FLUXNET (Pastorello et al., 2020), ICOS (RI, 2022), or SAPFLUXNET (Poyatos et al., 2021). These networks cover multiple biomes and multiple years or decades for some sites, offering unique insights about carbon-water fluxes and ecosystem functioning. However, these networks are relatively sparse and predominantly concentrated in North American and European regions (Pastorello et al., 2020; Poyatos et al., 2021).

As sap flow is driven by gradients in water potential within the vegetation, diurnal sap flow variations are closely coupled with the dynamics of vegetation water storage, considered here as vegetation water content (VWC), a quantity that can be estimated using microwave remote sensing e.g. (Konings et al., 2021; Saatchi and Moghaddam, 2000; Bernardino et al., 2024; Choudhury and Tucker, 1987). The most common approach is to estimate VWC vegetation optical depth, estimates of which have been obtained from many spaceborne microwave sensors (e.g. Frappart et al., 2020; Zotta et al., 2024; Konings et al., 2021). Current and planned microwave missions provide one snapshot every few days, observing "slow" ecosystem dynamics. They are adequate to observe inter- and intra-annual variations of above ground biomass (AGB), the slow response in water status over weeks and months, and to map (a-posteriori) biomass loss due to deforestation or mortality. However, this is not sufficient to capture the hysteresis illustrated in Figure 1 that can provide insight into plant water dynamics, health and stress.

This study is motivated by the idea to use sub-daily synthetic aperture radar (SAR) to observe vegetation health and stress (e.g. Steele-Dunne et al. (2024); Matar et al. (2024)). Many studies have shown that radar observations are influenced by dynamics in VWC. Dawn/dusk differences have been observed in spaceborne radar backscatter from vegetated areas and attributed to variations in water status (Steele-Dunne et al., 2012; Frolking et al., 2011; Friesen, 2008; Konings et al., 2017). Daily cycles obtained by aggregating Ku-band backscatter exhibit water loss/recharge patterns analogous to those illustrated in Figure 1 (Paget et al., 2016; Emmerik et al., 2017; Konings et al., 2017; Prigent et al., 2022). Sub-daily variations in radar observables (backscattering coefficient, coherence, phase) have been observed using tower-based radars in a range of vegetation types and conditions (Ho Tong Minh et al., 2013; Hamadi et al., 2014; Ulander et al., 2019; Ulander and Monteith,

105

2022; Monteith and Ulander, 2022; Ouaadi et al., 2021; Mcdonald et al., 1990; Chakir et al., 2021; Vermunt et al., 2021, 2022; Khabbazan et al., 2022), with several of the more recent studies demonstrating the link to VWC using destructive sampling, and the link to SF using co-located in-situ sensors. This study will contribute to the development of such a mission by providing insight into the number and timing of observations needed to capture the hysteresis illustrated in Figure 1.

In this study, we investigate the sub-daily variability in sap flow and its link to diverse climate stressors, to understand whether high temporal resolution information on vegetation water content and fluxes can be used to identify early signs of plant stress. For this, we make use of sap flow measurements from the SAPFLUXNET database (Poyatos et al., 2021) at high temporal resolution (hourly) across a wide range of biomes. We test the degree to which it is possible to capture the key characteristics of hydraulic hysteresis with targeted sub-daily measurements using experiments characterizing hysteresis in SF with plausible data acquisition strategies, which is important for understanding the minimum requirements for observation frequency if observing sub-daily hydraulic dynamics with satellites.

Our analysis is guided by two central research questions: (1) Can plant stress be detected from the shape and dynamics of the daily sap flow cycle? (2) How often, and at what times, would sparse observations be sufficient to capture these stress signals? In order to address these questions, we first introduce daily metrics that describe the hysteresis in the diurnal sap flow response to atmospheric water demand. Then, we examine the seasonal and interannual evolution of these metrics at selected sites to understand how they vary in response to hydrometeorological conditions and stress. Finally, we compare several

observation scenarios to investigate how well these metrics could be determined with sparser data from satellite remote sensing.

## 2 Material and Methods

#### 2.1 Data and Study Sites

The SAPFLUXNET database (Poyatos et al., 2021) provides harmonized sap flow data from forest stands across 202 sites across the globe, covering the period 1995 to 2018. The sites are primarily located in Europe and the USA, with additional sites in Australia and South Africa, and a few in the tropics and high latitude regions. Each site offers half hourly or hourly time series, with an average duration of three years (Figure 2).

The database includes sap flow data expressed at different levels - plant, per sapwood area, or per leaf area - which are derived from heat dissipation, heat pulse and heat balance methods (Poyatos et al., 2021). Sensors are typically inserted into the sapwood at the trunk, providing a sap flux density, i.e., the rate of flow per unit sapwood area. Plant-level data can then be calculated by multiplying this density by the total sapwood area of the tree. To obtain leaf-level data, the plant-level sap flow is divided by the total leaf area, yielding an estimate of transpiration at the leaf scale. Not all site specific datasets include measurements for all three levels simultaneously. In this study, we selected only sites with plant-level data because it represents the largest scale of sap flow measurement, making it particularly useful for understanding whole-plant water dynamics (Poyatos et al., 2021).

In addition to sap flow (SF), the database includes observations of hydrometeorological variables such as photosynthetically active portion of incoming radiation (PAR, quantified as PPFD), 2m surface air temperature (TAir), precipitation (PRE-

**Figure 2.** Sites of the SAPFLUXNET database where hourly SF and VPD (hysteresis data), as well as air temperature (TAir) alone, or air temperature and shallow water content (SWC) have been measured at 80% of days with a length greater than 12 hours for at least 2 years. The colors indicate which additional climate variables are available: gray show sites with only TAir, reddish colors show sites with TAir and SWC. The size of the hexagons indicates the number of years where time series of the sap flow data and the hydrometeorological drivers overlap. The annotated sites are the northernmost site with valid temperature and SWC data (RUS\_POG\_VAR), the site closest to the equator (GUF\_GUY\_GUY) and the site with the longest timeseries of SF and VPD (FRA\_PUE).

CIP), volumetric top (0-30 cm) soil moisture (TSM) or vapour pressure deficit (VPD). With these ancillary variables, the SAPFLUXNET database allows the study of rapid tree responses to environmental conditions and helps to bridge the gap between ecosystem flux networks and remote sensing.

For the broader analysis of sub-daily sap flow dynamics and response to climatic drivers, we identified 33 sites where plant-level SF, as well as VPD, TAir and TSM are available. Specifically, we selected sites where days with these data overlapped for at least 80% of the growing season (defined based on day length > 12 hours and TAir > 5 °C) across two or more years. These 33 sites represent less than 20% of all sites in the SAPFLUXNET database. Figure A1 shows how the mean temperature and precipitation varies across the 33 study sites.

#### 2.1.1 Case studies

140

We selected three focal sites (highlighted in Figure 2) representing a range of climates from dry Mediterranean to tropical and cold permafrost environments (see Figure A1) for a more detailed analysis. The selected sites are a mediterranean forest

in Puéchabon, France (FRA\_PUE), a humid tropical rainforest in French Guiana (GUF\_GUY\_GUY), and a forest-steppe in Pogorelsky Bor, Russia (RUS\_POG\_VAR). FRA\_PUE was chosen for its exceptionally long time series of sap flow (SF) and vapor pressure deficit (VPD), providing a robust dataset for temporal analysis. GUF\_GUY\_GUY was selected as it is the site closest to the equator, representing a low-latitude humid tropical environment with a seasonal drought. RUS\_POG\_VAR represents the northernmost site, making it valuable for studying plant-water relations in cold, high-latitude conditions.

FRA\_PUE is located 35 km northwest of Montpellier in southern France. This region is characterized by hot, sunny summers with low rainfall and strong winds, which are alternated by cool, wet winters, with rainfall primarily occurring between September and April (Limousin et al., 2009). The nearby Mediterranean Sea moderates the climate, resulting in an average temperature of 15.5 °C. However, rising temperatures due to climate change will increase drought severity in the region (Limousin et al., 2010a; Misson et al., 2011). The forest was historically managed as a coppice, meaning that all trees are approximately the same age and belong to the same species. The evergreen holm oaks (*Quercus ilex*), which reach a height of less than 7m and a diameter of less than 13cm, are situated on a plateau in a limestone karstic plateau. The soil is a shallow silty clay loam filling the cracks in the karst, with a volumetric fraction of rocks above 0.75. This leads to rapid infiltration in the subsoil and a reduced soil water holding capacity. The 25 studied holm oak trees exhibit a typical seasonal sap flow cycle, with reduced Gross Primary Production (GPP) and transpiration during periods of drought (Limousin et al., 2009; Cicuéndez et al., 2015; ALLARD et al., 2008). This site has the longest set of measurements in the SAPFLUXNET data, spanning 15 years (2000-2015) of which we can use 2003-2015, since VPD is not available earlier.

The GUF\_GUY\_GUY site is located in French Guyana on the east coast of the Amazon rainforest, near the equator. The northernmost part of the Guyana Plateau is characterized by small, elliptical hills rising from  $10\,\mathrm{m}$  to  $40\,\mathrm{m}$  above sea level, all covered by over  $400\,\mathrm{ha}$  of undisturbed tropical rainforest. Tropical ecosystems in this region are marked by a humid climate, high temperatures with minimal seasonal variation, and nutrient-poor acrisol soils. The entire region experiences a dry season, most pronounced in September and October, due to changes in the Intertropical Convergence Zone (ITCZ). Despite this variability, the ecosystem maintains high transpiration rates from trees of various species, with an average height of 35 meters (Bonal et al., 2008; Aguilos et al., 2018). At this site we use data from 6 trees of mixed tropical species of varying heights from 2014 until 2016. The estimated age of the trees is 200 years.

RUS\_POG\_VAR is the most northerly site in the network with valid temperature and SWC data. It is located in the Krasnoyarsk forest-steppe zone in southern Russia, within the forest of Pogorelsky Bor (Urban et al., 2019). This region is characterized by extreme seasonal temperature variations and low rainfall throughout the year. Precipitation is lower during the dark winter months, when sunshine is limited. Winters tend to register sub-freezing temperatures (Figure 3). Warmer temperature in summer and increasing atmospheric water demand lead to high variation in atmospheric humidity, along with higher precipitation and transpiration rates. The soil is mainly composed of sandy loamy gray material, characterized by distinct layering, where the upper layers undergo intense leaching. (Barchenkov et al., 2023). Overall, evapotranspiration exceeds precipitation in this region (Urban et al., 2019), with groundwater stored at great depths. Root-zone soil moisture is the main limiting factor for radial increment in the warm season (July-August), and can result in water stress, when surface storage is depleted (Barchenkov et al., 2023). The experimental site in Pogorelsky Bor has an average age of 50 years during the growing season

in 2015 and 2016 and features a mix of six deciduous larch and three evergreen pine trees. Since larches favor wetter habitats, they respond more strongly to increased water stress and are expected to be replaced by pine species in the future (Urban et al., 2019; Tchebakova et al., 2023; Barchenkov et al., 2023). One of the key differences between the two species is their ability to regulate sap flow during periods of increased atmospheric demand, with the deciduous larches exhibiting weaker stomatal regulation under higher water stress conditions (Barchenkov et al., 2023; Tchebakova et al., 2023).

#### 185 **2.2 Methods**

190

200

Sap flow (SF) represents the rate at which a volume of water moves through the tree per second and can be considered a proxy for transpiration and, thus, often serves as an indicator of carbon assimilation. SF has a clear diurnal cycle, driven by variations in radiation and atmospheric water demand, and regulated by plant functioning. Here, we analyse the relationship between sub-daily SF and VPD, the partial pressure deficit relative to saturated conditions, an indicator for atmospheric dryness and water demand (Figure 1).

For each of the 33 sites, we aggregated half-hourly VPD and average SF data across all measured trees into hourly values per site. Since some sites only provided hourly data, this step ensured consistency across sites. We then analyse the diurnal cycle of paired SF–VPD values, as shown in Figure 3 (left column). As discussed in the introduction, we hypothesise that the sub-daily dynamics can be used to understand plant stress, and derive two metrics at a daily basis to characterize the diurnal SF–VPD relationship, illustrated in Figure 3 (left column): the magnitude of the morning slope of the curve (SLOPE) and the area of the curve (AREA).

The SLOPE is calculated as the coefficient from a linear regression of VPD and SF between the times of minimum VPD and maximum SF. AREA corresponds to the area enclosed by the hysteresis curve of SF and VPD during the diurnal cycle, measured as the area of a polygon formed by the observation points. The variable SLOPE shows outliers (upper 5 %) and sometimes negative values, particularly when VPD gradients throughout the day are too small, e.g., in winter. These values were excluded from the analysis, resulting in a filtered SLOPE. In order to make the values comparable across sites, we standardize the filtered SLOPE values with the median and scale the AREA to 0–1 per site. These quantities are indicated as sSLOPE and sAREA.

We analyse the seasonal variability of the two metrics for the selected sites along with the seasonal changes in PPFD used as a measure of solar radiation, TAir andTSM for the three focal sites. For this, we first evaluate the seasonal cycle of each variable per site (Figures 3) and in a second step correlate the seasonal cycles of SLOPE and AREA to each of the hydrometeorological drivers, as shown in Figures 4, A2.

Since we want to evaluate plant functioning and water fluxes under environmental stress, we analyse combined extremes of TAir and TSM (upper and lower 20 % of the data). We obtain the following four extreme conditions: cold & wet, hot & wet, hot & dry and cold & dry. If one of these clusters has less than eight samples, we discard it from the further analysis. Within each of the remaining clusters, we calculate the values of sSLOPE and sAREA and the hysteresis of the mean diurnal cycle (Figure 5).

Finally, to evaluate the feasibility of large-scale monitoring of plant stress, for example, through remote sensing of VWC (see 1), we investigate how these two sub-daily metrics depend on the temporal resolution of the sub-daily observations. For this, we tested four scenarios of sub-daily sampling: 3-hourly, starting at mid-night (8 times per day, 8TPD), 6-hourly, starting at mid-night (4TPD) and 6-hourly with mid-night or noon left out (3TPDday, 3TPDnight). We then derived the curves of SF–VPD at the coarser temporal resolution (Figure 6) and calculated the corresponding sub-daily metrics. To evaluate whether the metrics could still be reliably estimated at coarser temporal resolution, we determined the corresponding coefficients of determination between the metrics derived from the resampled time series and the metrics derived from the hourly data for the complete data record at each site.

#### 3 Results

## 3.1 Sub-daily and seasonal dynamcis of SF-VPD and hydrometeorological drivers

#### 3.1.1 Seasonality of sub-daily hysteresis captured by daily metrics

The mean diurnal cycle of the bivariate relationship between SF and VPD for each site is shown in Figure 3 (left column), together with the corresponding metrics, AREA and SLOPE. We find that the diurnal hysteresis, characterised by the AREA of the hysteresis curves, and the morning sensitivity of SF to VPD vary over the course of the year at all sites, as vegetation responds to the evolving hydrometeorological conditions. This is illustrated by the seasonal cycles of daily values of SLOPE and AREA and the climate drivers (PPFD, TSM, TAir: Figure 3, right column) for each of the focal sites (FRA\_PUE, GUY\_GUY, RUS\_POG\_VAR).

First we discuss the diurnal dynamics of SF and VPD relationships (Figure 3, left panel) across sites. Generally, SF and VPD show very low (close to zero) values during night-time, with SF increasing as VPD increases through the morning. SF peaks around peak VPD, before declining through the afternoon into the evening, in response to decreasing VPD. Hysteresis is therefore characterized by lower SF rates in the afternoon compared to periods in the morning with comparable values of VPD (Song et al., 2015; Zheng et al., 2014) as a consequence of a change in the velocity of the SF response to atmospheric demand in the either morning or afternoon or both. The absolute rates of SF, as well as the peak times of SF and VPD vary across sites, as does the degree of hysteresis in a typical cycle, reflecting differences in hydraulic supply and transport across vegetation types and site conditions. To explore potential links between the metrics and climatic drivers associated with plant stress, we examine the average seasonal cycles of daily values characterizing the diurnal cycle. Their correlations are discussed in Section 3.1.2.

At FRA\_PUE (upper panel), the average diurnal cycle shows strong hysteresis with the peak SF around solar noon before the peak VPD around 2pm. On average, the morning SF is aligned with rising VPD, which is captured by SLOPE. The rate of increase of SF with VPD begins to drop off late in the morning (10am), and remains depressed relative to decreasing VPD throughout the afternoon, leading to a characteristic hysteresis loop. The seasonal cycle of the morning SLOPE closely tracks the temporal evolution of TSM (Pearson's correlation, R = 0.60). In April, SLOPE values are are approx.  $0.5 \ m^3/(s \cdot kPa)$ ,

Figure 3. Left column: Illustration of the approach used to calculate the sub-daily metrics used in this study, based on the mean diurnal cycle of sub-daily VPD and SF measurements, aggregated over all trees and all days for the whole time span of available data for three selected sites: FRA\_PUE, GUF\_GUY\_GUY, RUS\_POG\_VAR. The morning slope (SLOPE) is calculated through a linear regression of all points (colored blue) between minimum VPD and maximum SF, while the area (AREA) is the area within the polygon obtained by linking data points at all available sub-daily timesteps. Right column: Seasonal dynamics of photosynthetically active portion of incoming radiation (PAR, quantified as PPFD), air temperature (TAir) and top soil moisture (TSM), AREA and SLOPE with interquartile ranges showing the range of the hydrometeorological drivers and the derived metrics per day across years. The vertical lines indicate the selected growing season, i.e. when the daylength is higher than 12 hours and temperature > 5 °C. Absolute values of SLOPE vary according to the magnitude of tree-specific SF rates.

260

decreasing slowly through late Spring and more abruptly with the decline in available soil moisture during the summer months, where morning slopes close to  $0 m^3/(s \cdot kPa)$  indicate strongly limited hydraulic transport. SLOPE gradually recovers through the Autumn as TSM recharges. In contrast, the AREA values follow more closely the seasonality of PPFD (R = 0.84), peaking in June, before declining as water limitation suppresses the SLOPE of the morning limb of the hysteresis curve. The interannual variability in SLOPE and AREA, given by the spread around the mean values, is generally greatest outside the core of the dry season.

Similarly to FRA\_PUE, the average diurnal cycle also shows a hysteresis at GUF\_GUY\_GUY (mid panel), but here, hysteresis appears to emerge as a result of stomatal restrictions on SF initiating earlier in the morning (breakpoint at 9am). Timings of peak SF and VPD align at 12pm and the afternoon SF is aligned with VPD, with a strong stagnation at night close to minimum SF until SF starts to rise again in the morning. Despite these differences, the morning slope of the hysteresis curve similarly responds to changes in top soil moisture (R = 0.68), with slopes of 20-40  $m^3/(s \cdot kPa)$  during the wettest part of the year (January-July), decreasing to a minimum of approx. 8  $m^3/(s \cdot kPa)$  in November, when soils are driest. AREA is closely related to PPFD (R = 0.64), but as observed at FRA\_PUE, AREA starts to decrease in the driest parts of the year when SLOPE is suppressed by lower soil moisture.

In contrast, at RUS\_POG\_VAR, the average diurnal cycle is aligned with the diurnal cycle of VPD with very little hysteresis (lower panel), with peaks at 3pm and minima at night. Variations in the AREA of the diurnal hysteresis curve through the growing season (May/June-September) are also strongly and positively correlated with temperature (R = 0.71) and PPFD (R = 0.66). However, SLOPE is variable throughout this period, and less closely correlated with soil moisture than at other sites (R = 0.14). Instead SLOPE is related to temperature in the cold climate (R = 0.46), similar to AREA.

#### 3.1.2 Relation of SLOPE and AREA to climate drivers

**Figure 4.** Heatmap of Pearson's correlation coefficients of the seasonal cycles of the metrics SLOPE and AREA and hydrometeorological drivers (TSM, TAir and PPFD). Results are shown for each of the focal sites individually and the results for all sites. Maps of coefficients at all sites are provided in Supplementary Materials (Figure A2).

We analyse the relationship of SLOPE and AREA metrics and hydrometeorological drivers (TAir, PPFD, and TSM) through the correlation of daily values at the three selected sites, shown in Figure 4 and across all sites (Figure A2).

For the three selected sites, both FRA\_PUE and RUS\_POG\_VAR show strong and comparable in magnitude correlations between AREA and TAir and AREA and PPFD. At the tropical site GUF\_GUY\_GUY AREA shows a much weaker relationship with temperature (R=0.37) than with PPFD (R=0.64). At the Russian site, the positive correlation of AREA with PPFD (R=0.66) and TAir (R=0.71) is much stronger than any other pair at this site, indicating that the hysteresis is more strongly connected to atmospheric drivers than soil characteristics.

The relationship of SLOPE with hydrometeorological drivers varies more across sites than for AREA. In general, the tropical and Mediterranean sites show similar patterns, with increasing SLOPE associated with increasing TSM (correlation values higher or equal to 0.6) and decreasing SLOPE with increasing TAir and PPFD (correlations lower than -0.31). In contrast, the Russian site exhibits an inverted pattern, which results in both metrics responding in the same direction to the drivers. The observation, that SLOPE decreases with higher TSM is contradictory and will be discussed later. However, the coefficients at this site are generally less strong. At FRA\_PUE, the negative relation of SLOPE to PPFD (-0.31) is almost half of its relation to temperature (-0.56)

Comparing the correlations of SLOPE and AREA across the global collection of sites (N=33, 4 ("all"), A2) highlights considerable variability in relationships between sub-daily hysteresis characteristics and environmental drivers. AREA is generally positively associated with TAir and PPFD (mean R values of 0.60 and 0.61 across all sites, respectively), and negatively associated with TSM (mean R=-0.24), although the strength of these correlation varies across sites (Figure A2). For most sites, no clear pattern emerges by latitude, though there may be a climate-related pattern. The relationship between SLOPE and environmental drivers is less consistent across the global network of sites, resulting in very low average correlation values (Figure 4), although sites with a positive correlation of the metrics to TAir and PPFD exhibit a negative correlation to TSM, and vice versa (Figure A2). Interestingly, some sites show a close-to-zero relation of SLOPE to all three drivers, namely TSM, PPFD, and TAir.

#### 3.2 Diurnal SF-VPD dynamics during extreme conditions

To compare diurnal SF–VPD dynamics under varying climatic conditions, we cluster days according to extreme percentiles of temperature and top soil moisture. This procedure yields four clusters (cold & wet, hot & wet, hot & dry and cold & dry), although not all are present at all sites since we discard clusters with fewer than eight samples. The clusters and their corresponding mean diurnal cycles of absolute SF-VPD are shown in Figure 5. Given that we selected the three sites based on their different climate zones, the values of Tair and TSM differ strongly in their absolute ranges for the different clusters. Nevertheless, this allows to compare relative extremes in hydrometeorological drivers, and the corresponding response of SF to VPD.

Across the three sites, and despite their different background climate, sub-daily dynamics exhibits similar responses to local hydrometeorological extremes (Figure 5). Hot days, associated with higher VPD values, are generally characterized by higher AREA values than cold days at all three sites. Hot and dry days tend to be associated with a reduction in the morning SLOPE,

indicating suppression of transpiration driven water loss, even under very high VPD, when soil moisture is most likely to be a limiting factor.

Site-specific responses refine this general picture, even if absolute values of SLOPE vary across sites due to the magnitude of SF. FRA\_PUE shows the clearest separation between hydrometeorological regimes: cold—wet days combine low AREA with high SLOPE, hot—wet days show high AREA with reduced SLOPE and the highest peak SF, and hot—dry days reveal negative SLOPE anomalies with lower AREA than hot—wet days but still higher than cold—wet days. At this site, nighttime VPD minima do not return to zero under hot conditions, although SF can still cease. GUF\_GUY\_GUY show distinct dynamics for high-and low-TSM regimes, although it is worth noting that dry days are still considerably wetter than dry days in FRA\_PUE and RUS\_POG\_VAR. SLOPE remains similar across the wet regimes, but AREA increases in hot days, though the difference of hot versus cold is only 5 °C at this site. At hot-dry days, peak SF exceeds all other regimes by more than a factor of two. In RUS\_POG\_VAR, in contrast, SF-VPD dynamics separates mainly by temperature. An optimal range begins at about 8 °C, with high AREA occurring only at the warm end. Peak SF is reached on hot—dry days, whereas cold days show tight alignment between SF and VPD. On hot—dry days, SF stagnates slightly at mid-day but SLOPE remains high, a clear contrast to the warmer sites.

**Figure 5.** Upper row: Scatterplots of absolute TAir vs. TSM values for each focal site. Extreme percentile clusters (upper and lower 20 % of TAir and TSM) are colored by sSLOPE and scaled by sAREA; clusters with fewer than eight samples are not shown. Lower row: Diurnal cycles of SF (y-axis) vs. VPD (x-axis) for the identified clusters, with SLOPE shown in blue and AREA indicated by black circles with different linestyles for the hydrometeorological regimes: hot&wet (solid), hot&dry (dotted), cold&dry (dashdotted), cold&wet (dashed).

Finally, we find specific breakpoint of the average diurnal cycle per cluster, which highlight the diversity of site responses (Figure 5). Breakpoints in the diurnal SF—VPD relationship mark the onset of stomatal regulation, when atmospheric water demand through VPD keeps increasing, but SF increase slows down, or SF remains constant or even decreases. FRA\_PUE exhibits inverting breakpoints during hot—wet days, consistent with onset of strong stomatal regulation during the morning before VPD reaches its maximum early in the afternoon. GUF\_GUY\_GUY also shows a morning breakpoint for hot-wet days, but SF increases at a slower rate with increasing VPD, rather than inverting. The hot-wet regime generally occurs only rarely in RUS\_POG\_VAR, underscoring the climatic constraints of this energy-limited site.

In hot-dry days all 3 sites show a morning breakpoint, which is followed by a stagnation of SF under increasing VPD for both FRA\_PUE and GUF\_GUY\_GUY, and a slow-down of the SF increase in RUS\_POG\_VAR. Together with a typical stagnation to nearly constant nighttime SF during hot—wet days at the warmer sites in this regime, this increases AREA of the hysteresis curve. Both FRA\_PUE and GUF\_GUY\_GUY feature afternoon breakpoints at similar or lower VPD thresholds compared to the morning breakpoints, especially for hot-wet conditions.

Cold conditions tend to be associated with smaller VPD ranges and lower hysteresis across the three sites, with a breakpoint in peak SF very close to peak VPD for both cold-wet and col-dry conditions. In FRA\_PUE and GUF\_GUY\_GUY, cold and wet conditions show the highest value of morning slope. The cold-dry extremes are only found at RUS\_POG\_VAR, where SF\_VPD curves exhibit small hysteresis. This regime occurs at the very start and end of the growing season, where overnight temperatures are likely to be below freezing, and there are potential interactions between hydraulic transport, snow and frost.

## 3.3 Dependence of sub-daily metrics on sampling rate

The analysis thus far has been based on sap flow, continuous observations of which can be obtained from in-situ sensors. The fine temporal resolution ensures that the diurnal cycle is well-captured and the SLOPE and AREA can be calculated with some accuracy. However, what if fewer data are available? The limitation of sap flow data is that they are available only for few sites, and covering relatively short periods. Other datasets, for example, from remote sensing could in principle address this limitation and provide global coverage, but trade-offs in temporal resolution usually need to be considered.

While in situ sensors can provide continuous sap flow data, we consider the potential to estimate these descriptors of the diurnal cycle using temporally sparse data. In particular, as sap flow is connected to changes in water storage, which can be estimated using microwave remote sensing, we examine the degree to which the SLOPE and AREA can be estimated for several acquisition strategies with different numbers of observations and acquisition times. To determine optimal measurement frequency in order to derive the two metrics discussed above, e.g. using remote-sensing observations or more temporally sparse measurements, we analyzed the coefficients of determination for the metrics AREA and SLOPE across varying sampling frequencies for the longterm period of each site. We tested frequencies ranging from 8 (3-hourly), 4 (6-hourly), and 3 (6-hourly with either mid-night or mid-day left out) times per day (TPD) starting at 6am, as illustrated in Figure 6 (left panel) for FRA\_PUE. Each box summarizes the distribution of coefficients of determination obtained from correlating the daily values of the two variables for each of the 33 sites, thus reflecting the overall variability and strength across the 33 sites.

Figure 6. A) Diurnal cycle (average across days and trees) of measurements in FRA\_PUE at different sample rates: the gray cycle in the background of each subplot represents hourly observations. The black overlaid polygons show observations at four different sample rates, which are used to calculate the metrics from these lower sample rates. B) Boxplots of coefficients of determination ( $R^2$ ) between daily time series of the metrics at different sample rates (x-axis) and the hourly reference. One  $R^2$  was calculated per site for the whole period of available data, so the distributions summarize the variability of correlations across all sites. These values mainly reflect how closely the seasonal dynamics of the metrics from resampled time series of SF and VPD track those of the hourly reference across sites, shown for AREA (black) and SLOPE (blue).

The synchronization of resampled and hourly AREA is generally well preserved across sampling strategies, particularly at higher temporal resolution, and declining performance at reduced sample rates. With 8TPD, the metrics are well preserved across all years, with a median annual coefficient of determination of 0.95 and an interquartile range of 0.9–0.97. For 4TPD, the relationships between the metrics derived from sparse sampling and the higher temporal resolution become weaker and less well constrained, as given by the median correlation of 0.76 and interquartile range of 0.67–0.83. Sampling at 3TPD results in further degradation, with median values of R2 of 0.3 and 0.74 for 3TPDnight and 3TPDday, respectively, and corresponding interquartile ranges of 0.22–0.45 and 0.66–0.83.

Compared to AREA, the estimates of SLOPE are more sensitive to sampling frequency and exhibit greater variability across the sites, especially at 8TPD. At 8TPD, the correlation with the hourly sampling shows a median value of 0.5 with interquartile range of 0.26–0.82. Similarly to the performance of AREA, the agreement of SLOPE calculated with lower sampling rates deteriorates with decreasing frequency: with median correlations of 0.16, 0.01 and 0.14 for 4TPD, 3TPDnight and 3TPDday, respectively. Also the correlations of SLOPE show large variability across sites for 4TPD and 3TPDday, with interquartile ranges of 0.03–0.55 and 0.02–0.37, respectively.

The diurnal cycles of SF–VPD under sparse sampling, shown in the left panel of Figure 6, illustrate the poor performance of 3TPDnight in estimating SLOPE at least for FRA\_PUE. 3TPDnight cannot accurately capture maximum SF (see Figure 6, left), even though it does effectively capture minimum VPD. In contrast, 4TPD and 3TPDday slightly overestimate minimum VPD but still yield better SLOPE estimates, with 3TPDday performing best among the reduced sample rates. In general, AREA is underestimated at lower sampling frequencies. Likewise, maximum VPD tends to be underestimated across all sample rates, reducing accuracy in metrics relying on its diurnal extremes at this site. At other sites the peak times of SF and VPD and the general evolution of the diurnal cycle differ from FRA\_PUE, potentially leading to different best sample rates. Nevertheless the samplerates of 4TPD and 3TPDday still capture the metrics similarly at all sites, with better results of AREA at 4TPD and better results for SLOPE at 3TPDday.

#### 4 Discussion

# 4.1 Metrics as indicators for tree stress

The diurnal cycle of SF and VPD provides diagnostic indicators that consistently discriminate the drivers of extreme stress across three contrasting forested sites. Sap flow (SF) alone is not a reliable stress indicator, because absolute rates depend on tree size, species, and stand conditions (Wheeler et al., 2023; Chen et al., 2012), as also seen by the large differences in absolute SF across the three selected sites (Figure 5). Instead, we suggest that diurnal SF–VPD dynamics provide clearer insight into regulation processes. Our results show that the morning SLOPE and the AREA of the diurnal cycle of SF-VPD vary across seasons and systematically change from cold–wet to hot–dry conditions. Our results indicate that the combination of higher climatic demand and lower soil water availability reduce the coupling of SF to VPD.

We have shown that under cold and wet conditions, the diurnal cycles show high SLOPE and low AREA, as SF aligns tightly with VPD and hysteresis is minimal. This dynamics indicates a strong coupling of SF and VPD, i.e., under conditions

of limited demand and high supply, trees exhibit weak stomatal regulation, prioritizing carbon uptake over preventing water losses. By contrast, in hot and dry conditions, AREA increases and the SLOPE declines, reflecting the onset of regulation of SF before peak VPD and stagnation of SF during periods of high atmospheric demand. These patterns may result from increased stomatal regulation or limitations to plant hydraulic water supply.

Despite these general patterns, the variations of SLOPE and AREA and their relationships to hydrometeorological drivers differ among climate zones (Figures 4, A2). The full exploration of the complexity of the correlations between metrics and climate drivers is beyond the scope of this analysis. Furthermore, different responses of the metrics were found across sites, but still they help to reveal whether down-regulation occurs under increasing atmospheric demand. The underlying cause of down-regulation, be it stress, adaptive behavior, or species-specific traits—must be interpreted in context. Next to site-specific differences, the still short SAPFLUXNET time series limit the calibration of the metrics against long-term means. Without such references, it remains uncertain whether regulation corresponds to stress or adaptive behavior. Importantly, a lack of regulation does not necessarily indicate optimal conditions; it may instead signal failure to respond to stress or overshooting, risking excessive water loss and hydraulic damage (Schymanski et al., 2013; Lawson et al., 2011; Jones et al., 2022; Sperry and Love, 2015).

Nevertheless, the metrics derived from the SF–VPD relationship leave identifiable fingerprints of tree functioning under changing climates. They serve as early indicators of regulation patterns and water use strategies, some of which can be attributed to stress (Gambetta et al., 2020; Bodner et al., 2015; Brunner et al., 2015; Seleiman et al., 2021; Acosta-Motos et al., 2017). While SLOPE measures the strength of the coupling between SF and atmospheric demand, the AREA is an indicator of regulation timing in relationship to the atmospheric demand. Combining SLOPE and AREA helps to reduce ambiguity, as in the diagnostic case of low SLOPE with high AREA, which can be used as an indicator for strong regulation. Still, limitations remain: coarse temporal resolution may miss the identification of downregulation from breakpoint patterns in morning SF, and lack of regulation over consecutive days may reflect either risky strategies or optimal conditions. Ultimately, interpreting these metrics requires integration with local water balance, species traits, and climatic drivers. Together, the consistent finding is that downregulation of SF in response to rising VPD reflects a fundamental trade-off: trees reduce carbon uptake to conserve water. While the meaning of this trade-off varies by site and condition, the derived metrics reliably capture its imprint in the diurnal SF–VPD hysteresis.

## 405 4.2 Evaluation of the state of ecosystems in focus

More detailed assessment of the hydraulic hysteresis reveals novel insights into the variation in physiological dynamics across sites, with implications for understanding their adaptation to existing climate and potential vulnerability to future climate extremes.

At the Mediterranean site, stomatal regulation is triggered when atmospheric and soil moisture drought coincide (3), reflecting interactions between water and temperature, through VPD. The sensitivity of AREA and SLOPE to both atmospheric
demand and TSM is shown by the sign and strength of the correlation with all hydrometeorological drivers (4). Furthermore,
the negative correlation of AREA with TSM indicates that stomatal closure is a primary response when topsoil water becomes

a limiting factor. Increasing AREA and the occurrence of an inverting morning breakpoint (5) signals moderate regulation at high temperature, while declining SLOPE throughout the day represents strong conservative water use under severe diurnal drought and heat stress, triggering stomatal closure. A strong ability to close stomata is also indicated, when SF is absent at night at non-zero VPD. This site also has the widest range in TAir and TSM, which are as well negatively correlated, suggesting that plants frequently operate near or at their physiological limits. While the adaptation strategies have traditionally allowed these species to withstand seasonal drought, the additional compounding stress from increasing heat and VPD leaves them vulnerable to conditions that exceed historical variability. Thus, the site's long-term water conservation strategies, although beneficial in historical contexts, now signal a heightened vulnerability to future extreme events, and the empirical evidence of recent mortality supports the conclusion that these adaptive strategies are being overcome under novel climatic stresses (Peguero-Pina et al., 2020; Allen et al., 2015; Aurelle et al., 2022).

In contrast, stomatal regulation at the tropical site is less sensitive to heat. Here, the SF–VPD dynamics is mostly controlled by radiation and soil moisture (4), where seasonal drought results in the decreasing of the morning SLOPE. AREA primarily reflects afternoon or nighttime stagnation of SF, which can also be a sign for high TSM enabling the trees to maintain SF during the night (5). Previous studies have shown the strong importance of VPD for SF dynamics in these regions (Horna et al., 2011; Maréchaux et al., 2018; Suárez et al., 2021), here we show that radiation still contributes to modulate the sensitivity of SF to VPD, as seen by the strong correlations between AREA and SLOPE with PPFD and top soil moisture (Figure 4). At this site, tree height and hydraulic structure likely contribute to the high absolute SF rates, supported by deep rooting and access to water beyond topsoil layers (Horna et al., 2011; Apgaua et al., 2015; Kotowska et al., 2021; Spanner et al., 2022). The results highlight an ecosystem optimized for heat under conditions of high water supply. Trees at this site are capable of transpiring large volumes of water to maintain levels of high stomatal conductance and photosynthesis.

The Russian site displays an inverted response pattern compared to the warm sites: SLOPE and AREA increase and decrease together at the beginning of the season until they are aligned (3), which is supported by the strong correlation of AREA-PPFD and AREA-TAir (4). The cluster analysis (5) indicates that the trees at this site are predominantly energy-limited, with an optimal temperature threshold, which is likely to be occur earlier within a season with rising temperatures (Berner et al., 2013). Contrary to the other two sites, SLOPE decreases with increasing TSM, thus contradicting the expectation that additional water would sustain SF. This suggests unusual or site-specific regulation dynamics, potentially influenced by melting early in the season, growth and senescence of the larch canopy, or the fact that the seasonal drought is often outside of the growing season. The overall absence of strong stomatal downregulation at this site suggests that trees follow a non-conservative water-use strategy, possibly at the risk of depleting water for more carbon uptake. However, this may mask delayed stress onset or increased vulnerability to prolonged warming (Liu et al., 2022; Urban et al., 2017; Ruehr et al., 2019, 2015).

Projected climate change will likely intensify the stressors already identified here. In Mediterranean regions, increased frequency and duration of hot–dry extremes may amplify regulation and risk hydraulic failure, pointing to high vulnerability. In tropical forests, warming and drying may shift regulation triggers, potentially exposing hidden hydraulic limitations despite deep water access. At cold continental sites, continued warming may lift energy constraints, increasing SF without corresponding regulation—yet such non-conservative strategies may conceal longer-term risks of water imbalance.

## 4.3 Metrics and Sample Rates

Current research is actively exploring the relationship between VOD and various indicators of plant water status, contributing to a more in-depth understanding of plant health and ecosystem dynamics (Schneebeli et al. (2011), Momen et al. (2017), Holtzman et al. (2021), Humphrey and Frankenberg (2023), Asgarimehr et al. (2024)). Here, we analysed the effect of temporal resolution on the two metrics proposed. This allows to evaluate, in cases where 30-min or hourly resolution is not feasible, e.g., for remote-sensing applications, what the minimum temporal resolution would be required in order to adequately capture the SF–VPD dynamics. Generally, no sample rate can capture all details of the diurnal cycle as constrained by the hourly data (6). The 8TPD represents an ideal case assuming unlimited observations or coverage, consistent with characteristic frequencies from geostationary satellites for example. While time series from measurements at 8TPD exhibit very good correlation with the reference from hourly observations of SF and VPD for AREA, the results for SLOPE already show a degradation of the signal, with a broad spread across sites. For lower sample rates, results for both metrics are less accurate than for 8TPD, with the 75% percentile for SLOPE already falling below 0.6 at 4TPD. Compared to 4TPD, the correlations of 3TPDday for SLOPE demonstrate only slight differences, since only nighttime is excluded, where sap usually does not flow significantly. Therefore,

measurements at 3TPDnight, which include only nighttime flux, would not be a suitable choice for both metrics.

The mixed correlation results of SLOPE from resampled SF and VPD are likely due to the low signal-to-noise ratio of the hourly metric itself. These mixed results originate in dynamics resulting in negative SLOPE or high positive and negative outliers, such as rainy days or dew, which are hard to interpret. It is possible that SLOPE calculated from SF and VPD at lower sample rates could better infer the signal needed for early warning of tree stress hotspots by neglecting the values of SF and VPD that produce outliers. Another potential reason for the low accuracy of SLOPE from resampled observations is the possibility of both under- and overestimation, depending on the hysteretic behavior of the diurnal cycle, which shifts the maximum value of VPD behind the maximum of SF. Furthermore, the peak times of SF and VPD at some sites (e.g. the russian site) did not match solar noon. While the dynamics at the proposed times and sample rates would be well captured at FRA\_PUE and GUF\_GUY\_GUY, both metrics of the diurnal cycle could be underestimated at sites with dynamics similar to RUS\_POG\_VAR.

These results raise the question of whether the choice of metrics is adequate to capture the diurnal cycle of SF and VPD from satellites and how they can be improved. AREA is generally underestimated, but relative trends in AREA tend to be consistently retrieved with coarser temporal resolution. Even if resampled SLOPE has lower coefficients of determination than AREA, it indicates water limitation at most sites. Nevertheless, alternative approaches should be evaluated. For example, instead of the absolute slope, the ratio of the actual to a potential morning regression slope could be considered. This metric could identify collapse days, when the ratio is high, and exclude outliers when it exceeds 1. The challenge here mainly lies in how a potential slope could be defined.

Furthermore, the relationship between sub-daily VOD and SF velocity data needs to be evaluated to determine the optimal final sample rates. The reduced variance of the metrics under high environmental stress conditions indicates a stabilization of underlying processes and enhances the predictability of stress patterns during these periods. Therefore, SLOPE and AREA

can detect early warning signs of SF regulation, which indicates tree health, from the sub-daily response of SF to VPD, which could then be further investigated in detail through in situ observations to enhance our knowledge of specific stress responses outside of laboratory settings and mitigate subsequent tree mortality events. For this purpose, a sample rate of 3TPD centered at day-time, would suffice.

#### 4.4 Linking fluxes to storage

One limitation of the current study is the reliance on sap flow data (a flux), while the quantity more likely to be provided by satellite remote sensing is VOD (or derived VWC), a measure of vegetation water storage.

being considered. Sap flow measurements quantify the movement of water through the sapwood of individual tress. Although sensing depth of VOD and its sensitivity to water content in different compartments of the vegetation are known to depend on frequency, VOD (and derived VWC or biomass) are commonly assumed to represent storage of the vegetation as a whole, and spatially integrated across some footprint. Therefore, the dynamics observed in sap flow data, and the derived daily slope and area metrics should not be considered a trivial proxy for VOD or a change in VOD. That said, sap flow data reveal when transpiration is occurring which allows us to identify when, and the degree to which storage is likely to be changing at sub-daily scales. While it seems logical that steeper slopes and larger areas in sap flow would also result in similar changes in VOD, this should be confirmed with sub-daily VOD estimates from tower-based radars and/or radiometers and emerging GNSS-related techniques.

#### 5 Conclusions

510

The aim of this study was to understand how sub-daily sap flow dynamics can be used as an indicator of vegetation functioning and stress. While sap flow alone has limited interpretability in assessing vegetation condition, its response to sub-daily atmospheric water demand variations can help identify different types of stress. We propose two metrics of the diurnal cycle, the SLOPE, describing the SF sensitivity to VPD during the morning and the AREA describing the magnitude of diurnal hysteresis, that show clear seasonal and interannual variability patterns, associated with varying temperature and water availability conditions. As interpretion of either metric alone can be ambiguous, it is recommended that these metrics should be interpreted in combination.

Our results show that the short-term dynamics in SF response to atmospheric water demand can reflect long-term responses to stressors, emphasizing the potential for observations of sub-daily vegetation water dynamics to monitor ecosystem vitality at larger scales. Given the lack of global measurements of SF or vegetation water content at high temporal resolution from which to derive such metrics, we analyse their dependence on temporal sampling. Decreasing the sampling rate to 6-hourly at 4 times per day or even 3 times per day is sufficient to derive these two metrics and retain some of the information of the finer temporal resolution measurement. However the morning SLOPE requires finer sampling than the AREA, which indicates hysteresis strength. Nevertheless, our results show potential for temporally and spatially continuous measurements, e.g., from

future satellite missions, to use sub-daily information about vegetation water content and fluxes to track plant stress as an early-warning indicator for mortality events.

Therefore, enhanced vegetation monitoring through high-resolution satellite observations of vegetation water content could greatly improve ecosystem management strategies and climate change mitigation plans.

Code availability. The code can be made publicly available upon publication.

Data availability. SAPFLUXNET database (Poyatos et al., 2019)

## 520 Appendix A

#### $\mathbf{A1}$

**Figure A1.** Mean precipitation and mean air temperature at each of the 33 study sites. Symbol size scales with median sAREA and color indicates median sSLOPE per site across years with the three selected sites annotated and highlighted in red.

**Figure A2.** Maps of Pearson's correlation coefficients between the seasonal cycles of SLOPE and AREA and those of the hydrometeorological drivers (TSM, TAir, PPFD) at each site. These correlations illustrate the climatic factors associated with the seasonal evolution of the metrics.

525

Author contributions. AB, SSD, and AS designed the study. AS conducted the analysis with scientific input from AB, SSD, and DM. AS wrote the first draft of the paper and prepared all figures, including the conceptual representation. JML contributed site expertise and additional data from the FRA\_PUE site, which shaped the idea. All authors contributed to manuscript revision and approved the submitted version.

Competing interests. The authors declare that they have no conflict of interest.

Acknowledgements. The work was supported by the European Space Agency (SLAINTE, 4000139242/22/NL/SD).

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
