# Peer review of "Vegetation health monitoring based on sub-daily sap flow variability"

_EGUsphere, 2025_

## Referee Comment (RC1)

**Reviewer comments:**

**General comments**

In their manuscript "Vegetation health monitoring based on sub-daily sap flow variability", Tschackow et al. explore whether high-temporal-resolution sap flow data and its relationship to climatic variables can serve as an early indicator of plant stress. Using SAPFLUXNET data, they link sub-daily sap flow variability to environmental drivers. While the research question is interesting, the presented results only partially support the stated objectives. My main concerns are outlined below.

- 1) The rationale for selecting the three focus sites is not sufficiently justified. Beyond identifying patterns absent in the broader dataset, it remains unclear how these sites advance the overarching objective. The connection between the analyses based on the three sites and the global dataset is weak. The authors note (Lines 282–283) the absence of consistent patterns across latitudes, yet proceed to discuss relationships observed only at the three selected sites. This raises concerns about the validity and generalizability of these interpretations—particularly since some sites exhibit near-zero relationships between SLOPE and the key drivers (TSM, PPFD, and Tair). As the study aims to identify patterns that could inform remote sensing—based upscaling of vegetation health monitoring, the lack of consistency across the broader dataset suggests that this objective has not been achieved. To strengthen the manuscript, the authors could consider focusing exclusively on the three detailed sites and clearly state that the conclusions are site-specific. While this would limit the potential for upscaling, it would ensure that the conclusions are well supported by the data.
- 2) The distinction of timescales at which responses are assessed is not very clear: sometimes there is no mention of timescale—which can confuse the reader— and sometimes daily and diurnal are mentioned interchangeably. Often the sub-daily and diel processes are not explained distinctly from daily processes.
- 3) The concepts of "stress" and "vegetation health" are insufficiently defined. It is unclear how the authors determine that exceeding a certain area or threshold constitutes stress. This definition appears to lack physiological justification and is not validated against independent measurements of plant stress (e.g., water potential, leaf turgor, or critical hydraulic thresholds).
- 4) Previous work (e.g., Wan et al., 2023, Agricultural and Forest Meteorology) has shown that sap flow–VPD hysteresis is related to sapwood area, yet this factor is neither mentioned nor accounted for. Additionally, the temporal progression of hysteresis (e.g., clockwise vs. counterclockwise loops and their changes over time) is not analyzed. The discussion would benefit from a clearer linkage to prior studies on sap flow hysteresis and environmental drivers. Since hysteresis in sap flow is well documented, the novelty of this finding in the current manuscript remains limited. Without robust evidence that these relationships hold beyond the three case sites, the reliability of these proposed metrics for general vegetation monitoring remains uncertain.

5) Lastly, the conceptualization (e.g., Figure 1) does not include bimodal sapflow peak responses and thus excludes species that show a midday depression in sapflow due to high VPD (see for example Kumar et al. 2023 *Journal of Experimental Botany*). To ensure conceptual completeness and global applicability, the authors should revise their framework and discussion to incorporate such species-specific responses.

**Specific comments:**

- Title: vegetation health or vegetation water stress?
- Line 6: diurnal or sub-daily? Diurnal is not about sub-daily variations but variation from one day to another
- How is stress defined? And how is health and stress related? How do the authors define poor or good vegetation health? (at which stress level and why?)
- Not clear from the Abstract what species or climate region are the focus of the findings
- Line 10: soil moisture at what time scale? Daily? There is typically a lag between stem refilling at night. How do the authors then relate the instantaneous sapflow response to instantaneous soil moisture?
- Line 12: at what time scale?
- Line 17: sub-daily hysteresis?
- Is Figure 1 a graphical abstract? Otherwise, shouldn't come before the Introduction
- Figure 1 caption: what determines stress with respect to the hysteresis? Panel a does not describe the concept: What differentiates the different circles? The different curves and the satellites? What satellites are these? Each panel should be described clearly (especially there is no explanations of panels c and e). There is a mention of stress on the figure, but is this a specific event (e.g., point in time) or in general? What is then the conditions shown in the other panels where no stress is mentioned?
- Line 46: stomatal conductance is only one of the mechanisms that determine plant water uptake, the stem capacitance and refilling of water between day and night provides a buffer to whole plant water uptake which needs to be mentioned as well.
- Line 49 the statement needs a reference. See for example Zweifel at al. 2021 *New Phytologist.*
- Line 56: the caption does not explain that this is what the figure is showing.
- Line 57: be specific please which plant hydraulic functioning you mean?
- Line 80: SAPFLUXNET is not a running network like FLUXNET or ICOS. It provides a data product with no commitment to extend beyond 2018.

- Line 85: how do the authors link vegetation water content to sapflow where instantaneous water content is a product of sapflow and stem capacitance. There is no mention of the role of stem capacitance in the whole paper which is an essential component of the plant water uptake
- Figure 2: What is the range of years for different symbol sizes? How should the readers interpret the number of years from the size of symbols?
- Please explain what is the aim of selecting three focal points; what analysis here is different from the 33 study sites and why a subset of sites was needed? Were there additional info available at these three sites that were not available in other sites?
- Lines 150-184 please give consistent level of information for each site: MAT, MAP, elevation, species botanical names.
- Line 165: how high is "high temperature"?
- If possible, please report mean site LAI (any information about canopy cover is important for interpreting forest water use and if not directly measured, can be extracted from satellite products). Please also mention the DBH range for the trees used in the analysis for each site as the size of the trees is a determining factor.
- Case studies should come in the Methods section not before.
- Line 177: how deep is "great depths"?
- Line 179: by site age you mean age of the trees?
- Line 190: but Figure 1 includes also soil moisture and not only VPD. Please stay consistent with describing the objectives.
- Line 207: Figure 4, Figure A2
- Line 207: Please remove reference to result, in the Methods section
- Line 209: upper and lower 20 % of the which data? Were the extremes defined per site?
- Line 210: what is meant by each sample? And why eight?
- Line 214: typo in see 1 (should be see Figure 1?)
- Figure 3 caption should explain what TSM is.
- Figure 3: X-axis label is missing.
- Figure 3: In the lowest panel right, why is the slope and area missing for some days within the growing season?

- Figure 3 Left panel: please explain the caption what the vertical and horizontal dashed lines mark.
- Figure 3: be explicit please in the caption which variables, instead of writing hydrometeorological drivers.
- Figure 3: displayed PPFD is also diurnal? Please clarify in the caption by adding "daily" before each variable.
- Lines 224-239: this part reads as a mixture of a repetition of the introduction and what could potentially be the Discussion. Please first report on results.
- Lines 301-312: please indicate where the readers can see these results.
- Figure 4 please indicate on the matrix, if any of these correlations are not significant.
- Overlay of results on the global map, what is the added info to display these on a global map?
- Figure 5: are these slopes all significantly different from zero? Please indicate if they are, and remove those which are not.
- Why is it that the displayed VPD in Figure 5 reaches 3 and 2.5 kPa for the warmer sites, but maximum VPD in Figure 3 doesn't reach that maximum? Please also indicate if VPD and TSM in Figure 5 are mean daily values.
- Line 251 and 313: statistically how is the breakpoint determined? The information should be added to the Methods section.
- Lines 313-319: where do the readers see these results?
- Please show the breakpoints on the figures.
- When deriving the breakpoints, are they estimated based on all slopes, or only the statistically significant ones?
- Lines 331-345: text repeats from the Introduction/Methods, and does not belong to Results.
- Line 342: "daily" frequencies
- Line 345: of which two variables?
- Figure 6: caption mentions panels A and B, but these are not marked on the Figure
- Figure 6: x and y-axis values are missing
- Figure 6: the legend does not include all items that are displayed on the figure (e.g., the solid lines with different colors)
- Figure 5 and Line 344: why only FRA\_PUE? It is not clear.

- What is the hypothesis underlying the testing of whether r2 between hourly and daily metrics match?
- Line 373: the concept is not new and is well-established, to look at the SF-VPD hysteresis rather than absolute SF. The authors should at least refer to previous studies and discuss their findings in the context of existing knowledge.
- Lines 375-376: please indicate if this observation holds across all sites or only for a selection of sites, and then adjust the conclusion accordingly.
- Line 377: please indicate across which sites, since the study includes both the analysis of a selection of sites, and analysis of all sites.
- Lines 393-397: this repeats text from the Introduction.
- Line 401-404: Firstly, finding a hysteresis in sapflow data is not new (numerous studies show this) and whether it relates to soil moisture or VPD is not confirmed in this study, apart from on a selection of three sites. So I am not sure how reliable these metrics are for making such conclusions like this statement.
- Line 416: which site?
- Line 419: please provide a reference for this statement.
- Line 420: based on what results do the authors conclude a change in vulnerability over time?
- Line 421: what do you mean by "novel climatic stresses"?
- Line 424: Figure 4? There seems to be a typo.
- Line 426: typo? Figure 5? Same in line 434 and line 435.
- Line 438: "snow" melting?
- Line 440: what is the reference for the statement that says seasonal drought is often outside the growing season?
- Line 449: abbreviation (VOD) not explained yet

---

## Referee Comment (RC2)

**Reviewer Comments-**

**General Comments:**

Tschackow et al. explore high temporal sap flow data and its relationship with VPD, temperature, and topsoil moisture across three different sites. In general, the work shows promise but needs more exploration or refining the objectives based on the figures shown. My major concerns are 1) lack of rationale for selecting three sites, and why such different sites. This work would be more impactful if it includes more than one station per ecotone, which exists based on Figure 2. 2) Objectives are not explicitly clear, nor what is defined as stress or a hot/cold day.3) On line 14 the authors mention that the data is temporally sparse, which might lead to lack of alignment across data sets but do not tackle in the methods how corrections were made to account for that. 4) The writing would benefit of some revision, particularly on paragraph structure, correct citations, and improving clarity. For example, line 496 is a one-sentence 'paragraph' that does not relate to the rest of the conclusion.

**Specific comments:**

- o Definitely improve flow on the abstract, making sure transitions are clear.
  - More concise
  - There is also no clear distinction between the abstract and introduction.
- o Unclear how Figure 1 fits in the introduction. Seems more like an abstract
- On the introduction, there is no clear distinction between how rising CO2relates to plant water dynamics. I would suggest adding a sentence that links the paragraphs together
  - Please revise citations
    - Line 48- Foster has no year, is it a one author publication?
    - Line 50- no year
    - Line 71- no year
    - Line 187- requires citation
  - Line 76-78: The hypothesis is conceptually interesting but fundamentally circular, since hysteresis itself reflects stress; the authors should clarify which specific aspect of hysteresis they expect to change prior to observable canopy-level decline.
- Begin by mentioning sap flow acronym at the beginning, not at the middle of intro, and being consistent with the acronyms. If not, there is no need to include it, especially SF for sap flow
- Why contrasting sites? What is the motivation? Why not concurrent mediterranean sites, or concurrent tropical sites? Is three different sites without replication informative enough?
- Paragraph 80, yes and? How does it add to the flow?
- Figure 3- Align the mean seasonal gradient to the right panels, not the middle.
  - X axis?
  - Would also recommend better ID for the different sites. They are unnecessarily complicated

- Line 449- there is no justification to why Vegetation Optical Depth research can serve to motivate your work. It can be informative on its own. I would suggest restructuring towards how it can inform remote sensing initiatives.
  - Similar comment on line 496. If you want to use your work for VOD, then it is worth mentioning in the conclusion, but it does not add to your argument in any way
- There is no mention of how sap flow is measured (heat thermistor, etcare the sensor different, and does not cite literature depicting that SF has a clear diurnal cycle- it is not always the case and is very much individual dependent.
- o Line 177: great depths?
- Line 179: age of trees? Or age of the forest?
- No division between major groups such as angio/gymno clades, which might inform why there is similar patterns across the different sites.
- There is lack of consistency throughout the writing
- What is the broader application of this work? It is not explicit.